# Studying Corrosion Using Miniaturized Particle Attached Working Electrodes and the Nafion Membrane

**DOI:** 10.3390/mi12111414

**Published:** 2021-11-18

**Authors:** Jiyoung Son, Edgar C. Buck, Shawn L. Riechers, Shalini Tripathi, Lyndi E. Strange, Mark H. Engelhard, Xiao-Ying Yu

**Affiliations:** 1Energy and Environment Directorate, Pacific Northwest National Laboratory, Richland, WA 99354, USA; jiyoung.son@pnnl.gov (J.S.); edgar.buck@pnnl.gov (E.C.B.); shawn.riechers@pnnl.gov (S.L.R.); shalini.tripathi@pnnl.gov (S.T.); lyndi.strange@pnnl.gov (L.E.S.); 2Environmental Molecular Sciences Laboratory, Pacific Northwest National Laboratory, Richland, WA 99354, USA; mark.engelhard@pnnl.gov

**Keywords:** nanoparticle, working electrode, microfluidic electrochemical cell, electrochemical analysis, particle attached electrode, Nafion membrane, System for Analysis at the Liquid-Vacuum Interface (SALVI), CeO_2_

## Abstract

We developed a new approach to attach particles onto a conductive layer as a working electrode (WE) in a microfluidic electrochemical cell with three electrodes. Nafion, an efficient proton transfer molecule, is used to form a thin protection layer to secure particle electrodes. Spin coating is used to develop a thin and even layer of Nafion membrane. The effects of Nafion (5 wt% 20 wt%) and spinning rates were evaluated using multiple sets of replicates. The electrochemical performance of various devices was demonstrated. Additionally, the electrochemical performance of the devices is used to select and optimize fabrication conditions. The results show that a higher spinning rate and a lower Nafion concentration (5 wt%) induce a better performance, using cerium oxide (CeO_2_) particles as a testing model. The WE surfaces were characterized using atomic force microscopy (AFM), scanning electron microscopy-focused ion beam (SEM-FIB), time-of-flight secondary ion mass spectrometry (ToF-SIMS), and X-ray photoelectron spectroscopy (XPS). The comparison between the pristine and corroded WE surfaces shows that Nafion is redistributed after potential is applied. Our results verify that Nafion membrane offers a reliable means to secure particles onto electrodes. Furthermore, the electrochemical performance is reliable and reproducible. Thus, this approach provides a new way to study more complex and challenging particles, such as uranium oxide, in the future.

## 1. Introduction

Cerium dioxide (CeO_2_) adopts a fluorite structure (space group Fm-3m) under ambient conditions, which is common to a variety of dioxides, including uranium oxide (UO_2_), thorium oxide (ThO_2_), plutonium oxide (PuO_2_), and doped zirconium dioxide (ZrO_2_) [1]. Therefore, CeO_2_ has been used as a surrogate to understand irradiated mixed oxide (MOX)-based matrix fuel due to its similarity in structure and chemical and mechanical properties [2]. For instance, CeO_2_ was used to study irradiated mixed oxide as irradiated fuel pellet materials in fission reactors due to its similar structural and mechanical properties [3]. CeO_2_ and UO_2_ can be engineered to feature similar grain sizes, and CeO_2_ is often used as an analogue of UO_2_ in dissolution studies [4]. When considering the safety case for the geological disposal of nuclear waste, CeO_2_ and ThO_2_ are effective spent nuclear fuel analogues that approximate the microstructure characteristics of fuel-grade UO_2_, yet they are not as sensitive to changes in the oxidation state of the cation as uranium [5].

Previous efforts utilized millimeter (mm)-sized UO_2_ spent fuel to reduce the risk of contamination to study the spent fuel corrosion phenomenon [6]. For example, Sunder et al. utilized sliced UO_2_ pieces that were a few mm laterally and 3 mm thick, and they successfully studied UO_2_ corrosion in an electrolyte containing hydrogen peroxide under open-circuit corrosion conditions [7]. Other previous efforts also used disc type bulk electrodes to achieve results [8,9,10]. However, such experiments are not easy to conduct; this is largely due to the difficulty of handling highly radioactive materials. Hot cells and other logistics were required to perform such experiments. Therefore, new methods are needed to make such studies more accessible to develop the predictability of spent fuel corrosion potential.

Microfluidics is a viable approach to address the associated technical challenges. Our team developed a microfluidic-based platform for the multimodal spectroscopy and microscopy of liquids [11,12,13]. This vacuum-compatible platform is named System for Analysis at the Liquid-Vacuum Interface (SALVI). The electrochemical SALVI, or E-cell, contains three electrodes and allows simultaneous electrochemical analysis, which enables multimodal spectroscopy and microscopy including chemical imaging tools such as time-of-flight secondary ion mass spectrometry (ToF-SIMS) and scanning electron microscopy (SEM) [14,15].

We recently introduced a working electrode (WE) fabrication method to include a small amount of nanoparticles in the SALVI cell using stamping [16]. Conductive epoxy was used as a medium for the direct attachment of nanoparticles onto the electrode surface by utilizing several stamping methods. It showed promise for spent fuel corrosion studies using electrochemistry. However, the inconsistency of stamped target particles and the difficulty in controlling the epoxy amount on fabricated WE call for improvement and optimization. A new approach using Nafion, a proton exchange membrane, is developed in this work. Nafion is a perfluorinated polymer with sulfonic acid groups that stands out for its high proton conductivity, its selective permeability to water, and its superior chemical stability [17,18,19]. Nafion also has been widely used in fuel cells and batteries as the separator and binder materials due to its high proton transfer ability. Specifically, Nafion is used as the barrier membrane material between anodic and cathodic layers in fuel cells [18,20,21,22]. Nafion is used to hold the target particles (i.e., cerium oxide CeO_2_) onto the gold (Au) conductive substrate to form a precisely controlled WE in terms of mass loading and electrode surface area.

In this study, we demonstrate that using micrometer (μm)-sized electrodes is a novel and alternative solution to study particle corrosion. This is achieved by miniaturizing the working electrode in an established electrochemical cell, that is, the electrochemical version of the SALVI, or the E-cell [11]. The Nafion membrane is used as a protection layer in the E-cell after attaching CeO_2_ particles. Considering the challenges involved in utilizing UO_2_, a radioactive chemical, as a WE component with regard to strict safety protocols of radioactive material handling [14,15], its close analogue CeO_2_ was chosen to develop and optimize the fabrication protocol.

The fabricated CeO_2_-containing SALVI E-cell devices were tested to show reproducibility and to validate their performance as a new analytical approach. Multimodal surface analysis tools, including time-of-flight secondary ion mass spectrometry (ToF-SIMS), scanning electron microscopy-focused ion beam (SEM-FIB), X-ray photoelectron spectroscopy (XPS), and nanometer resolution atomic force microscopy (AFM) were used to characterize the powder-attached electrode surface to verify the electrically driven corrosion effects on the electrode surface. Microfluidics provides the benefit of singling out the α-radiolysis effect due to the small size of the overall electrochemical cell as an inherent feature. The novel microfluidic approach increases the degree of freedom to study more diverse spent fuel corrosion conditions, including synfuels and noble metal particles, in the future.

## 2. Materials and Methods

### 2.1. Chemical Agents

The Nafion solutions of 5 wt% and 20 wt% were acquired from Sigma Aldrich. The pristine single crystal CeO_2_ particles were provided by Dr. Dallas Reilly. The cerium oxide (CeO_2_, 10 nm mean diameter) particles were purchased from US Research Materials Inc. (Houston, TX, USA). The polydimethyl siloxane (PDMS) was purchased from Millipore Sigma (Darmstadt, Germany). The silicon (Si) wafer chips (University Wafers, Boston, MA, USA) were used in device development. More details were described previously [16]. The platinum (Pt) wires (25 mm outside diameter) were purchased from Sigma Aldrich (St. Louis, MO, USA).

### 2.2. Microfluidic Electrochemical Cell Fabrication

#### 2.2.1. WE Fabrication

A LAURELL 650M spin coater (North Wales, PA, USA) was used to form a thin layer of Nafion with 0, 500 and 1000 rpm spinning recipes. The Nafion membrane was used to hold the target particles when making the WE that contained CeO_2_ particles. Figure 1a shows the WE fabrication process using Nafion as the protection membrane. A small amount of target CeO_2_ particles (e.g., 250 mg) was suspended in 1 mL of deionized (DI) water to make a stock with a concentration of 250 mg/mL, as shown in Figure 1a. A few droplets of the particle-suspended liquid mixture were pipetted onto the shadow-masked Au-WE layer on the Si chip [16]. The Si chips were used to reduce the cost of method development. The spun Nafion layer was cured in a 70 °C oven for another 30 min to form a homogeneously distributed layer adapted from a protocol in Nafion membrane fabrication [20,21]. The Si chips with deposited target particles were dried in the oven at 70 °C for 30 min. Optical images of each completed WE surface also were recorded with an optical microscope (VHX500, Keyence, Itasca, IL, USA) with a magnification 100 times (see Appendix A). It is worth noting that the optimal amount of particle deposition was investigated by testing many devices in duplicate sets. For example, the initial amount of CeO_2_ particles was set at 0.6 mg. The particle mass loading was increased to show the enhanced effect in the CV testing.

#### 2.2.2. Device Assembly

The SALVI E-cell uses Pt wires as the counter electrode and reference electrode, respectively. The body of the devices was fabricated using soft lithography [23,24]. More details on SALVI fabrication were reported previously [16]. Key details are provided in the Appendix A.

### 2.3. Membrane Thickness Mesurements

The Nafion membrane thickness was determined using a profilometer (Profilm3D, KLA, Milpitas, CA, USA).

### 2.4. Cyclic Voltammetry

Cyclic voltammetry (CV) was used to verify the microfluidics device performance. The electrolyte containing 0.1 mol/L of sodium perchlorate (NaClO_4_) was used following a previous publication using bulk electrodes [7]. A series of CV potential scans at different scanning rates was performed using multiple E-cells to verify the electrochemical reproducibility of this new technique. Figure 1b shows the experimental setup of the SALVI E-cell. The CeO_2_-containing E-cells were tested using CV sweeping from −1 V to 1 V and 1 V to −1 V at several scan rates (i.e., 10, 20, 40, 60, 80, and 100 mV/s), respectively. CV scans on the Nafion control SALVI E-cells were also performed to compare with those obtained from the devices with CeO_2_-particle WEs. The WE chips were retrieved for surface characterization using ToF-SIMS, AFM, and SEM-FIB. In comparison, the pristine, as-made WE chips containing CeO_2_ were characterized to determine the corrosion effects after CV sweeping.

### 2.5. AFM Characterization

A topographical analysis of the as-made and corroded WE surfaces was performed using an MFP-3D Infinity AFM (Asylum, Oxford, Santa Barbara, CA, USA). Tapping mode measurements were performed using an etched silicon probe (Bruker, RTESPA-300, 8 nm nominal tip radius, 40 N/m spring constant, Billerica, MA, USA) with a set point of 3 V and a scan speed of 0.3 Hz.

### 2.6. ToF-SIMS Characterization

A surface analysis of the as-fabricated WE surfaces was performed using a ToF-SIMS (IONTOF GmbH, ToF-SIMS V, Münster, Germany). The pressure of the main chamber was maintained at 1 × 10^−8^ mbar during the analysis. The primary ion beam was a 25 keV Bi_3_^+^ with 10 kHz pulse energy. The pulse width was 0.8 ns and the current was ~0.6 pA. ToF-SIMS high resolution spectral data were acquired by rastering over an area of 500 × 500 μm^2^ for 60 scans. The ToF-SIMS data were processed using the IONTOF Surface Lab 7.0 software. The mass spectra were calibrated using peaks such as C (m/z^+^ 12.0), CH_3_ (m/z^+^ 15.02), and Ce^+^(m/z^+^ 139.90) in the positive ion mode.

### 2.7. SEM-FIB

The SEM-FIB liftout samples were prepared using a FEI, Helios 660 (Thermo Fisher, Waltham, MA, USA). The lift-out specimens were obtained by milling out an interfacial sample of the CeO_2_ from the device. Milling was performed under the ion-beam source at 30 kV and 0.79 nA. The voltage was reduced to 5 kV for the final thinning of the cross-section. Electron micrographs of the pristine and corroded CeO_2_ electrodes were acquired using a secondary electron detector at 5 kV and 3.2 nA current.

### 2.8. XPS

The XPS measurements were performed with a Thermo Fisher NEXSA (Waltham, MA, USA). This system uses a focused monochromatic Al Kα (1468.7 eV) source for excitation with a double-focusing hemispherical analyzer with multi-element input lens and 128 channel detector. The X-ray beam is incident to normal to the sample and the photoelectron detector is 60° off normal. High-energy resolution spectra were collected using a pass-energy of 50 eV with a step size of 0.1 eV and a dwell time of 50 ms. The full-width-at-half-maximum (FWHM) was measured to be 0.82 eV for the Cu(2p_3/2_) peak using the same conditions as those at which the narrow scan spectra were collected.

The XPS Ce(3d) narrow scan region was deconvoluted using the 3d_5/2_ peaks in Casa XPS (v 2.3.24) with a GL (80) peak shape. The 3d_3/2_ peaks were not needed for analysis since they featured a fixed area ratio (2:3) and spin orbit coupling distances (18.10 eV). The charge referencing was performed using the most intense Ce 3d_5/2_ peak at 916.7 eV [25]. Additional fitting information can be found in the supporting experimental details in the section on Appendix A.

## 3. Results and Discussion

### 3.1. WE Fabrication Optimization

Two different Nafion concentrations (5 wt% and 20 wt%) of 10 µL droplets and three spinning speeds (i.e., 0, 500 rpm, 1000 rpm) were tested to reach an optimal result. When using different spinning rates, the Nafion membrane thickness varies. Table 1 presents a summary of the Nafion membrane thickness corresponding to the spinning rates. The fabricated Nafion layer using the optimized conditions is approximately 150 nm thick. After applying the series conditions to develop the Nafion coating on top of the target particles, an optimized condition was determined, that is, one drop of 10 µL of 5 wt% Nafion solution spin-coated at 1000 rpm. This device performance was verified by comparing the electrochemical analysis results of different devices prepared using the listed conditions in duplicates in Table 1.

Higher spinning rates result in thinner membranes and potentially better electrochemical results in the test cases evaluated in this work. The comparisons in Appendix A show distinct features of CV profiles such as peaks 0.3 V (from −1 V to 1 V) and 0.7 V (from 1 V to −1 V) from the Nafion control (Figure 2d). It is interesting that major peaks in Appendix A shift to the left by ~0.1 V, corresponding to the thinnest Nafion membrane in the 20 wt% Nafion batches of devices. This result indicates that the Nafion background in the CV profile becomes subservient as its layer getting thinner. Therefore, the optimal Nafion spin coating procedure would be a 1000 rpm spin using the 5 wt% Nafion solution. Additionally, increasing the target particle amount could increase the signal intensities to capture unique characteristics as well. An example is given in Appendix A.

To achieve optimal conditions, we chose to increase the particle mass loading by 10 times and to decrease the Nafion layer thickness, that is, increasing the CeO_2_ to 6 mg and using the thinnest Nafion from 5 wt% Nafion with a 1000 rpm spinning rate. Devices fabricated using this method show a significant improvement compared to those made using the conductive silver epoxy as the binder material, as described in our recent publication [16].

Using optimized conditions, the CeO_2_ device CV profile (Appendix A) results show that using the Nafion binder causes less interference and improves particle deposition in a more controlled manner. During potential sweeps of the oxidation process of the CeO_2_, the anodic peaks were observed at potentials (E_pa_) of −0.75 V and 0.022 V. In comparison, the Nafion control device’s CV profiles show two peaks at −0.596 V and 0.239 V, respectively. Similarly, during the reduction process of the CeO_2_, the observed cathodic peak potentials were (E_pc_) of 0.446 V and −0.134 V, respectively. By contrast, the Nafion control device CV profiles were 0.647 V, 0.056 V, and −0.476 V, respectively. We also plotted current vs. square root of scan rates to show linear trend relations in Appendix A. This finding of the CeO_2_-loaded E-cell demonstrates the good reproducibility of the devices using the newly developed Nafion fabrication method.

### 3.2. SEM-FIB Characterization

FIB-SEM was used to obtain images of the surface of CeO_2_ WE. Figure 2b shows the morphology of the CeO_2_ WE after 100 cycles of CV scanning. Corrosion makes the surface relatively rougher compared to the pristine CeO_2_ surface (Figure 2e). This observation can be explained by the possible redistribution of Nafion as a result of CV sweeping [17,26]. Figure 2c also shows a slight Nafion layer thickness reduction compared to Figure 2f, as indicated by the white marks. Additional SEM-FIB trench images are presented in Appendix A.

### 3.3. ToF-SIMS Spectral Analysis and 2D Mapping

The ToF-SIMS molecular mapping of the CeO_2_ particles as the WE in the SALVI E-cells and nanometer resolution AFM imaging of the electrode surface topography before (RMS: 140 nm) and after running CV (RMS: 428 nm) are depicted in Figure 3. Boththe ToF-SIMS and the AFM imaging results show that an even Nafion layer was deposited on top of the CeO_2_ particles before applying potential sweeps. The Ce^+^ and CeO^+^ signal intensities are lower in the pristine electrode surface (Figure 3a,b) compared to those after running CV (Figure 3g,h). This is because the CeO_2_ was covered by the Nafion layer. In contrast, the Nafion fragment ions (e.g., C_4_F_5_^+^, C_4_F_7_^+^) show much higher ion intensities in the as-prepared electrode surface (Figure 3c,d) compared to after its use in the electrochemical experiments (Figure 3i,j). After performing electrochemical corrosion, higher Ce^+^ and CeO^+^ signals are observed (Figure 3g,h). More similar observations are reported in Appendix A. The AFM results show that an even Nafion film (Figure 3e,f) is formed using the spin coating technique. Both ToF-SIMS and AFM observations indicate that the Nafion membrane provides a safe protection layer with CeO_2_ particles as the WE. Once it had been used in the CV experiments, the Nafion was redistributed due to proton transfer (Figure 3i,k), exposing the CeO_2_ particles to the electrolyte. It is worth noting that one hundred cycles of potential sweeps were performed using the WE inside a SALVI cell before the ToF-SIMS and AFM surface mapping. Our results show that the devices prepared using the novel Nafion and particle attachment method can be used for extensive electrochemical testing.

Figure 4 depicts the positive ToF-SIMS spectra of the corroded and pristine CeO_2_ WE surface. The spectral results corroborate with the 2D mapping. Much stronger Nafion fragment peaks, such as C_4_F_5_^+^ m/z^+^ 142.99 (Figure 4a) and C_4_F_7_^+^ m/z^+^ 180.99 (Figure 4c), are prominent on the pristine electrode surface. The peaks representing CeO_2_ particles, such as Ce^+^ m/z^+^ 139.90 (Figure 4a) and CeO^+^ m/z^+^ 155.90 (Figure 4b), are also observable in the pristine surface. However, their counts increased after potential sweeping and corrosion develops on the electrode surface, as seen in Figure 4d,e, respectively. by contrast, the Nafion fragment peaks C_4_F_5_^+^ m/z^+^ 142.98 (Figure 4d), and C_4_F_7_^+^ m/z^+^ 180.99 (Figure 4f) reduced after corrosion occurred, providing additional evidence that the Nafion redistributes. This phenomenon is not surprising, and it has been reported in the past [17,19]. It is worth noting that the possible sulfur and fluorine contamination on the WE surface was studied using ToF-SIMS spectral analysis. The SIMS spectral results provide no evidence of sulfur or fluorine interference, as shown in Appendix A [27,28].

### 3.4. XPS Analysis

XPS narrow scan regions were used to determine the oxidation states of the pristine CeO_2_ WE (Figure 5b) and post-CV WE (Figure 5a). Wide scan regions are also shown in Appendix A. The Ce (3d_5/2_) was deconvoluted to distinguish the oxidation states present. After CV (Figure 5a), the Ce^4+^ oxidation state was dominant, which indicates that electrochemical oxidation of the Ce^3+^ to Ce^4+^ occurred as a result of the CV scan. This Ce^4+^ oxidation state was also confirmed by the increase in the intensity of 916.7 eV and the disappearance of shoulders present in the pristine electrode shown in Appendix A. Two oxidation states can be observed in the pristine CeO_2_ WE case, which was dominated by Ce^4+^. This was likely due to air oxidation during storage [29,30]. The calculated ratio between Ce^3+^ and Ce^4+^ in the pristine WE case are 29.7% and 70.3%, respectively.

Further evidence of Ce^3+^ is shown in Appendix A by the presence of shoulders and a lower intensity peak at 916.7 eV relative to after electrochemical cycling or fully oxidized state [25]. The peak positions and intensity ratios (Appendix A) also resemble the results reported by Paparazzo et al. [25]. The conversion of the Ce^3+^ oxidation state to Ce^4+^ after electrochemical cycling indicates the redox activity of the cerium particles on the electrode surface. Thus, the XPS results further confirm that Nafion-assisted CeO_2_ particle deposition as WE is viable.

Direct deposition on an electrode (DDE) is a widely used approach to making electrodes. In fact, powders have been applied using DDE. However, controlling the amount of material dispensed onto a surface is difficult in DDE [31,32]. The new Nafion and particle attachment method provides a better solution compared to DDE. More importantly, the new development presented here allows flexible electrode preparation using different kinds of particles that are compatible with the microfluidics platform. CeO_2_ is an important material for energy applications [33,34]. The development of this DDE WE to be used in microfluidic cells can provide a multimodal platform for rapid electrochemical testing, for example, in studying rare earth materials [35]. In this study, we used CeO_2_ as an analogue to establish the protocol for studying uranium oxide corrosion in the future [4]. It is anticipated that the developed approach will be adapted to study other, more challenging materials, such as UO_2_.

## 4. Conclusions

CeO_2_ was used as a surrogate of spent fuel UO_2_ particles to develop a new method to fabricate effective WE in the SALVI electrochemical cell. A layer of thin Nafion membrane was developed via spin coating to protect the target particles. This new method provides a valuable electrochemical analysis tool for studying materials and understanding their corrosion potentials in the future. The fabrication workflow was optimized. The effects of Nafion weight percentages and spinning rates on the Nafion membrane thickness were investigated. Multiple devices were prepared using varying parameters. Their electrochemical performance was evaluated using CV. Our results show that the device-to-device reproducibility is reliable. More importantly, the electrochemical characteristics in the CeO_2_-containing devices feature distinguished redox potentials that differ significantly from those of the Nafion controls. It is expected that this new solution will be used to study spent fuel redox chemistry, since it will offer a diverse platform to introduce syn fuels, noble metal particles, or controlled dopants as WEs. More systematic studies will be performed using microfluidics to study the spent fuel chemistry at the material interfaces in the future.

## Figures and Tables

**Figure 1 micromachines-12-01414-f001:**
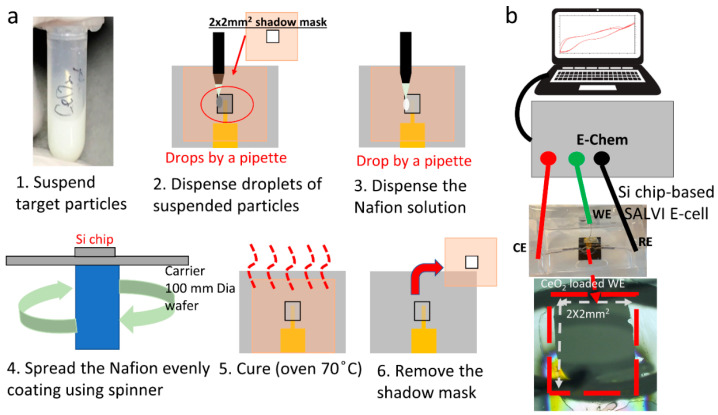
(**a**) The flow chart of the step-by-step fabrication process of particle deposition (1–3), Nafion film formation (4–5), and the assembled electrochemical device (6); and (**b**) the setup to perform electrochemical analysis of CeO_2_ containing SALVI E-cells.

**Figure 2 micromachines-12-01414-f002:**
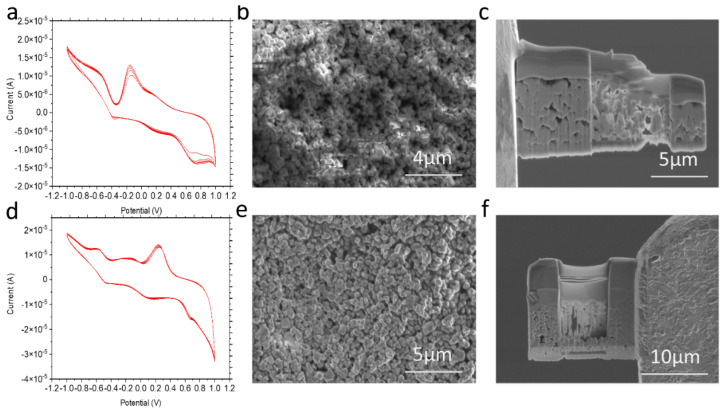
(**a**) CV results of CeO_2_-applied electrochemical cell using 5% Nafion, 1000 rpm spin rate and 80 mV/s scan rate, (**b**) SEM image of the CeO_2_ WE surface after electrochemical corrosion with a magnification of 14,000 X, (**c**) SEM-FIB image of the suspended lamella from the corroded CeO_2_ WE surface with a magnification of 12,000 X, (**d**) CV results of the Nafion only control using 5 wt% Nafion 500 rpm spin rate and 80 mV/s scan rate, (**e**) SEM image of the pristine CeO_2_ WE surface with a magnification of 12,000, and (**f**) SEM-FIB image of the suspended lamella from the pristine CeO_2_ WE surface with a magnification of 6500.

**Figure 3 micromachines-12-01414-f003:**
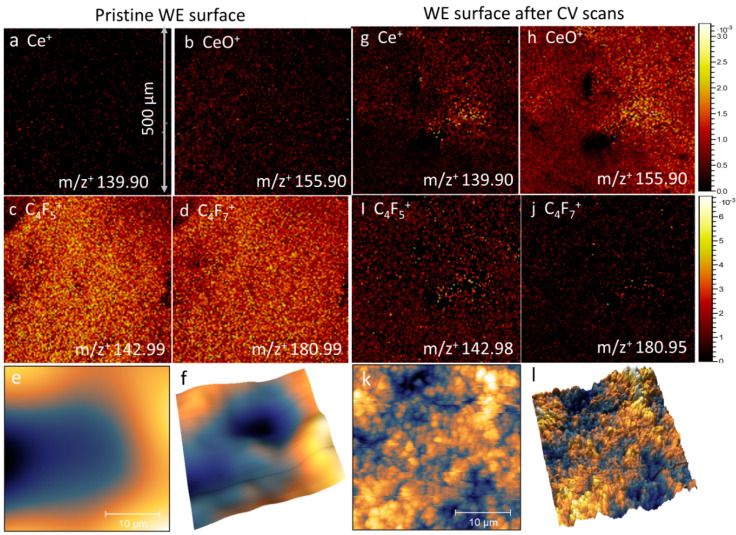
Normalized ToF-SIMS 2D positive ion images of representative CeO_2_ particles and Nafion fragments on a pristine CeO_2_ WE surface: (**a**) Ce^+^, (**b**) CeO^+^, (**c**) C_4_F_5_^+^, and (**d**) C_4_F_7_^+^. Corresponding AFM images of the as-fabricated electrode (**e**) 2D surface and (**f**) 3D amplitude images. Similarly, representative ToF-SIMS 2D positive ion images of the CeO_2_ WE surface after CV scans depicted in (**g**–**j**) and AFM 2D/3D images (**k**,**l**), respectively.

**Figure 4 micromachines-12-01414-f004:**
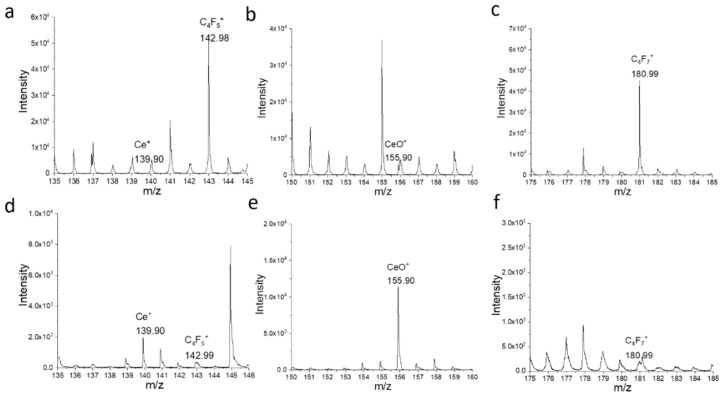
ToF-SIMS spectral comparison of (**a**–**c**) the pristine and (**d**–**f**) CV scanned CeO_2_ WE surface. Ce^+^ and CeO^+^ peaks are related to CeO_2_ while C_4_F_7_^+^ is related to the Nafion membrane.

**Figure 5 micromachines-12-01414-f005:**
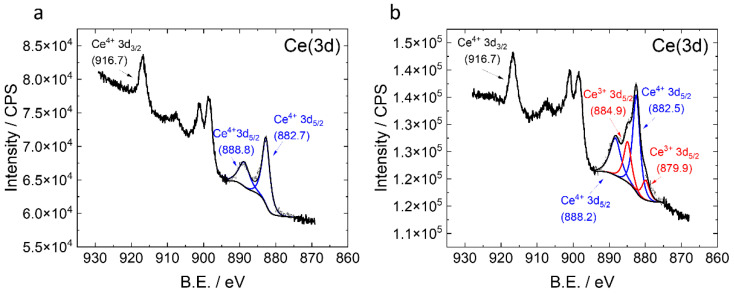
XPS narrow scan regions of the Ce(3d) after CV (**a**) and pristine WE condition (**b**).

**Table 1 micromachines-12-01414-t001:** Nafion membrane fabrication conditions for thickness optimization.

No.	Nafion (%)	Spin Rate (rpm)	Curing Time (min)	Footprint (mm × mm^2^)	Avg. Thickness (µm)	Note
1	5	500	30	2.1 × 2.2	0.13	Nafion only
2	5	1000	30	2.1 × 2.1	0.12	Nafion only
3	20	500	30	2.4 × 1.7	1.18	Nafion only
4	20	1000	30	2.0 × 2.1	0.83	Nafion only
5	20	500	30	2.5 × 2.4	14.21	Nafion over CeO_2_ deposit
6	20	1000	30	1.8 × 1.8	6.45	Nafion over CeO_2_ deposit

## Data Availability

Data is contained within the article or Appendix A.

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
