# Peer review of "Studying Corrosion Using Miniaturized Particle Attached Working Electrodes and the Nafion Membrane"

_micromachines, 2021, doi:10.3390/mi12111414_

Round 1

Reviewer 1 Report

The authors have prepared a significant supplementary part. If the journal allows (i.e. there is not a restriction in the length) it would be good to involve it into the submitted paper.

Special remarks

It would be good if the authors could emphasize better why the development of this multiple device is important, how could be used in the real life?

 Among the Keywords is mentioned “SALVI”. Please, give the meaning of the abbreviation here, too, though in one of the recent papers of the authors referred as [16] the reader could find it (System for Analysis at the Liquid-Vacuum Interface (SALVI)).

Page 2 line 8 from the bottom: “aka”.  It needs explanation.

Though for specialists in analytical chemistry the “NAFION” is well-known, but not all of the readers belong to this group. It would be useful to give some information about it (e.g. it is a sulfonated tetrafluoroethylene-based fluoropolymer co-polymer ionomer, proton conductor with good mechanical and thermal stability).

Conclusion: “vis” Not “via”?

Author Response

Author’s Reply for Reviewer #1

  1. It would be good if the authors could emphasize better why the development of this multiple device is important, how could be used in the real life?

Answer: We include a comment at the end of the results and added a discussion with a reference as follows,

“The development of this DDE WE to be used in microfluidic cells can provide a multimodal platform for rapid electrochemical testing, for example, in studying rare earth materials [32].”

  1. Among the Keywords is mentioned “SALVI”. Please, give the meaning of the abbreviation here, too, though in one of the recent papers of the authors referred as [16] the reader could find it (System for Analysis at the Liquid-Vacuum Interface (SALVI)).

Answer: Thanks! Revised.

  1. Page 2 line 8 from the bottom: “aka”.  It needs explanation.

Answer: Revised as follow,

“This is achieved by miniaturizing the working electrode in an established electrochemical cell, that is, the electrochemical version of the SALVI, or the E-cell [14, 15].” 

  1. Though for specialists in analytical chemistry the “NAFION” is well-known, but not all of the readers belong to this group. It would be useful to give some information about it (e.g. it is a sulfonated tetrafluoroethylene-based fluoropolymer co-polymer ionomer, proton conductor with good mechanical and thermal stability).

Answer: Thanks for your suggestion!  The following is added in the introduction,

“Nafion is a perfluorinated polymer with sulfonic acid groups that stands out for its high proton conductivity, selective permeability to water, and its superior chemical stability [17-19]. Nafion also has been widely used in fuel cells and batteries as the separator and binder materials due to its high proton transfer- ability. Specifically, Nafion is used as the barrier membrane material between anodic and cathodic layers in fuel cells [18, 20-22].”

  1. Conclusion: “vis” Not “via”?

Answer: The error has been fixed as suggested.

Reviewer 2 Report

The authors have provided a detailed and well-rounded presentation and characterization of a novel working electrode using Nafion. I believe the paper is written very well, giving in-dept details important for other researchers using various characterization techniques. The proposed development process is of high interest to a large community involved with nuclear fuel and its usage. The paper is of high quality as it provides all the necessary information, references and indications for the chosen practices of the authors.

I only have some small remarks, mostly related to typos or errors in referencing, which can be easily corrected:

In Supplementary image S5 you mention the speculated sulfur contamination not being present. Is this maybe another trait of this new system that is important? If so, you should definitely add this detail in the main manuscript text and reference it to the supplementary material. Otherwise maybe at least give some indication why this is important to provide in the supplementary information.

In section 2.1 :

“Cerium oxide (CeO2, 10 nm mean diameter) particles were purchased from the US research materials Inc. Chemicals were used as is.” Maybe write as: “Cerium oxide (CeO2, 10 nm mean diameter) particles were purchased from the US research materials Inc. Chemicals and used in the as-purchased state.”

“More details were described previously.” Please provide a reference to your previous work.

“2.1.1. WE fabrication” should be numbered with “2.2.1”

In section 2.2.1 :

“A LAURELL 650M spin coater was used to form a thin layer of Nafion with 0, 500 and 1000rpm spinning recipes.” Space missing between 1000 and rpm.

“The Si chips with de-posited target particles had been dried in the oven at 70 ºC for 30 min.” The degree symbol (°) should not be underlined.

“2.1.2. Device assembly” should be numbered with “2.2.2”

In section 2.2.2:

“More details on SALVI fabrication were reported previously.” Reference your previous research

In section 2.3:

“Figure 1B shows the experimental setup of the SALVI E-cell.” I assume that it should be Figure 1b.

In section 3.2:

“Corrosion makes the surface a relatively rougher compared to the pristine CeO2 surface (Figure 2e).” Omit the “a” between “surface” and “relatively”

“Figure 1c also shows a slight Nafion layer thickness reduction compared to Figure 1f as indicated in white marks.” I assume here the references are meant to be relate to Figure 2 and not Figure 1.

In section 3.3:

“More similar observations are reported in Figure S4.” The reference should be to Figure S5 and/or Figure S6.

In section 3.4:

“However, Controlling the amount of the material dispensed onto a surface is difficult in DDE [28, 29].” The word “Controlling” should not be capitalized.

Author Response

Author’s Reply for Reviewer #2

  1. In Supplementary image S5 you mention the speculated sulfur contamination not being present. Is this maybe another trait of this new system that is important? If so, you should definitely add this detail in the main manuscript text and reference it to the supplementary material. Otherwise maybe at least give some indication why this is important to provide in the supplementary information.

Answer: We included the following statement in the first paragraph of p8,

“It is worth noting that the possible sulfur and fluorine contamination on the WE surface was studied using ToF-SIMS spectral analysis.  The SIMS spectral results give no evidence of sulfur or fluorine interferences as shown in Figures S5S6.”

  1. In section 2.1 :“Cerium oxide (CeO2, 10 nm mean diameter) particles were purchased from the US research materials Inc. Chemicals were used as is.” Maybe write as: “Cerium oxide (CeO2, 10 nm mean diameter) particles were purchased from the US research materials Inc. Chemicals and used in the as-purchased state.”

Answer: As you suggested, we changed this sentence to “Chemicals were used in the as-purchased state.”

  1. “More details were described previously.” Please provide a reference to your previous work.

Answer: A reference has been provided, “More details on SALVI fabrication were reported previously [16].”

  1. “2.1.1. WE fabrication” should be numbered with “2.2.1”

Answer: Thanks! Corrected.

  1. In section 2.2.1 :

“A LAURELL 650M spin coater was used to form a thin layer of Nafion with 0, 500 and 1000rpm spinning recipes.” Space missing between 1000 and rpm.

Answer: Correction has been done as suggested.

  1. “The Si chips with de-posited target particles had been dried in the oven at 70 ºC for 30 min.” The degree symbol (°) should not be underlined.

Answer: Correction has been done as suggested.

  1. “2.1.2. Device assembly” should be numbered with “2.2.2”

Answer: Correction has been done as suggested.

  1. In section 2.2.2:

“More details on SALVI fabrication were reported previously.” Reference your previous research

Answer: A reference, [16], was added.

  1. In section 2.3: “Figure 1B shows the experimental setup of the SALVI E-cell.” I assume that it should be Figure 1b.

Answer: Correction has been done as suggested.

  1. In section 3.2: “Corrosion makes the surface a relatively rougher compared to the pristine CeO2 surface (Figure 2e).” Omit the “a” between “surface” and “relatively”

Answer: Correction has been done as suggested.

  1. “Figure 1c also shows a slight Nafion layer thickness reduction compared to Figure 1f as indicated in white marks.” I assume here the references are meant to be relate to Figure 2 and not Figure 1.

Answer: Thanks! The correction has been done as suggested.

  1. In section 3.3: “More similar observations are reported in Figure S4.” The reference should be to Figure S5 and/or Figure S6.

Answer: Correction has been done as suggested.

  1. In section 3.4: “However, Controlling the amount of the material dispensed onto a surface is difficult in DDE [28, 29].” The word “Controlling” should not be capitalized.

Answer:  Correction has been done as suggested.
